# Examining the surface evolution of LaTiO$_x$N$_y$ an oxynitride solar water splitting photocatalyst

Craig Lawley [1,2], Maarten Nachtegaal [1], Jochen Stahn [1], Vladimir Roddatis[3], Max Döbeli[4], Thomas J. Schmidt [1,2], Daniele Pergolesi [1✉] & Thomas Lippert[1,2,5✉]

LaTiO$_x$N$_y$ oxynitride thin films are employed to study the surface modifications at the solid-liquid interface that occur during photoelectrocatalytic water splitting. Neutron reflectometry and grazing incidence x-ray absorption spectroscopy were utilised to distinguish between the surface and bulk signals, with a surface sensitivity of 3 nm. Here we show, contrary to what is typically assumed, that the A cations are active sites that undergo oxidation at the surface as a consequence of the water splitting process. Whereas, the B cations undergo local disordering with the valence state remaining unchanged. This surface modification reduces the overall water splitting efficiency, but is suppressed when the oxynitride thin films are decorated with a co-catalyst. With this example we present the possibilities of surface sensitive studies using techniques capable of operando measurements in water, opening up new opportunities for applications to other materials and for surface sensitive, operando studies of the water splitting process.

[1] Paul Scherrer Institute, Forschungsstrasse 111, 5232 Villigen PSI, Switzerland. [2] Department of Chemistry and Applied Biosciences, ETH Zürich, 8093 Zürich, Switzerland. [3] Institute of Materials Physics, University of Goettingen, Friedrich-Hund-Platz 1, 37077 Goettingen, Germany. [4] Laboratory for Ion Beam Physics, ETH Zürich, 8093 Zürich, Switzerland. [5] Molecular Photoconversion Devices Division, International Institute for Carbon-Neutral Energy Research (I2CNER), Kyushu University, 744 Motooka, Fukuoka 819-0395, Japan. ✉email: daniele.pergolesi@psi.ch; Thomas.lippert@psi.ch

Water splitting using a photocatalyst and sunlight, where the light energy is converted into chemical energy stored in the form of $H_2$, is a promising route for the production of a sustainable, storable and transportable solar fuel, as the energy carrier to meet increasing global energy demands[1–3]. For this process, in semiconductor photocatalytic materials, electron/hole pairs are created upon absorption of sunlight. The photogenerated charges reach the solid−liquid interface at the surface of the semiconductor and are used for the water reduction and oxidation reactions.

Photochemical splitting of water into hydrogen and oxygen using titanium oxide ($TiO_2$) irradiated under ultraviolet (UV) light was first observed by Honda and Fujishima in 1972[4]. In the past several decades since the now termed Honda−Fujishima effect was first demonstrated, extensive research in this field has been targeted at trying to find suitable and efficient water-splitting photocatalysts on which the development of cost-effective and state-of-the-art technology could be based[5]. However, if the hydrogen evolution reaction does not represent a key limiting factor, the oxygen evolution reaction (OER) remains a major hurdle in the search of suitable materials[6], owing to the fact that the OER is a complex reaction severely hampered by sluggish kinetics and significant overpotential losses[7]. Meaningful progress has been made for oxide-based photocatalysts in furthering the insight into the reaction mechanisms, whilst also improving the performance and efficiencies of the photocatalysts[8,9]. However, these oxide-based materials are all wide band gap materials, sensitive only to the near-UV region of the solar spectrum, which consists of just a small percentage of its entirety, compared to the visible region which contributes to almost half[10]. Therefore, an alternative approach has been adopted related to the search of novel semiconductor photocatalysts with narrower band gap energies, which lie in the visible-light range, yet still possess band gap widths greater than the theoretical 1.23 eV required for the water-splitting process[11].

One promising class of alternative materials for visible-light-driven water splitting are perovskite oxynitrides, whose general formula can be written as $ABO_xN_y$ (where A = La, Ba, Sr, Ca; B = Ti, Ta, Nb; $x + y = 3$). The substitution of N into the O sites affects the position in energy of both band edges leading to a band gap width in the visible energy range (between 1.8 and 2.5 eV) with band edges comprising both, the hydrogen and oxygen evolution potentials[12–14]. There have been various studies to date regarding oxynitrides and they have shown to be effective photocatalysts for visible-light-driven water splitting[15–19]. However, there seems to be a common problem for oxynitrides in that degradation of the initial performance of these materials is generally observed. It is not uncommon to observe a 30−50% decrease in the initial performance during operating conditions[20]. To date, the vast volume of literature with respect to oxynitrides has predominantly comprised of the synthesis of oxynitride powders, improving synthetic routes and applying surface modifications to increase their photoelectrochemical (PEC) performance under varying experimental conditions[19,21]. The physiochemical characteristics of the oxynitride−liquid interface and its evolution during operation conditions has, to the best of our knowledge, had little investigation to date[22–24].

For this study, we aim to achieve a fundamental understanding of the physiochemical processes occurring at the oxynitride−liquid interface where the electrochemical reactions take place, which lead to degradation of the photocatalyst. We employ the complementary surface-sensitive techniques, neutron reflectometry (NR) and grazing incidence X-ray absorption spectroscopy (GIXAS) to probe the surface modifications of epitaxially grown $LaTiO_xN_y$ (LTON) films before and after PEC characterisation. We show that the combination of NR and GIXAS provides invaluable experimental approaches for surface-sensitive measurements (3 nm resolution) of the structural, electronic and compositional properties of materials. Importantly, these techniques are feasible with operando characterisations, where the experimental platform can be extended to the wider field where applicable.

## Results

**Oxynitride thin films.** The growth of oxynitride films using pulsed reactive crossed-beam laser ablation (PRCLA) described previously[25] allows the fabrication of thin films with different crystalline properties and tuneable nitrogen contents. Figure 1 shows the structural and morphological characterisations for LTON thin films grown for this work. The crystal structure for $LaTiO_2N$, shown in Fig. 1a, b, shows the X-ray diffraction (XRD) measurement for the epitaxial oxynitride LTON film deposited on titanium nitride (TiN)-coated magnesium oxide (MgO(001)) substrate. In our previous work[26] we demonstrated that the TiN seed layer can not only be used as a current collector for PEC measurements, but it also provides a very good template for the growth of oxynitride films making the deposition process more stable (controllable), whilst also increasing N content in comparison to oxide substrates.

TiN possesses the same rock salt structure as MgO and has similar lattice parameters, $a = 4.211$ Å (MgO), $a = 4.235$ Å (TiN). Due to the small lattice mismatch between MgO and TiN (ca. 0.56%), this layer grows (100) epitaxially oriented with MgO (001), the (200) diffraction peak is visible as a shoulder on the peak of the substrate. LTON has the orthorhombic perovskite structure with lattice parameters $a = 5.5731$ Å, $b = 7.8708$ Å, and $c = 5.6072$ Å.

The XRD pattern shows that LTON grows epitaxially on the TiN buffer layer with the ($h,k,l$) reflexes (0,2,0), (0,4,0) and (0,6,0) appearing at $2\theta$ values ca. 22, 46 and 72 respectively. The MgO substrate (200) reflex has been marked in green. The compositions of the thin films used in this work were determined by the RBS/ERDA measurements (Supplementary Fig. 1 and Supplementary Table 1). The compositions of the LTON films consist of La:Ti and O:N ratios of 1.04−1.06 and 10.3−10.7 respectively. The experimental uncertainties on the compositions measured by RBS for La and Ti and ERDA for O and N are ±2% and ±3% respectively. We assume that the chemical composition of the films, while affecting the light absorption and charge migration properties, has little influence on the evolution of the physico-chemical state of the surface layer.

Figure 1c shows the transmission electron microscope (TEM) cross-sectional image of the epitaxially grown film, with the selected area electron diffraction (SAED) pattern inset in the top left corner. The TEM image indicates a well-defined surface and interfaces between the layers. The SAED pattern also confirms the epitaxial growth of LTON and TiN on the MgO substrate. The protection layer seen in the cross-sectional image was applied on the sample solely for the TEM measurement. The high-angle annular dark-field (HAADF) image with atomic resolution is shown in Fig. 1d. The O:N ratio for the LTON films was determined using electron dispersive spectroscopy (EDX) and are in agreement with the RBS/ERDA analysis. Although the O:N ratio is low, this is usual for epitaxially grown LTON films where there is a trade-off between crystalline quality and total nitrogen substitution/content. Films grown using larger laser fluence tend to grow polycrystalline films with much higher nitrogen contents.

**Experimental strategy.** To study the physicochemical evolution of the surface of the semiconductor, the use of oxynitride thin films as model systems provides a clear advantage over the corresponding powders in the fact that they offer relatively wide, atomically flat, and well-defined surface where one can more

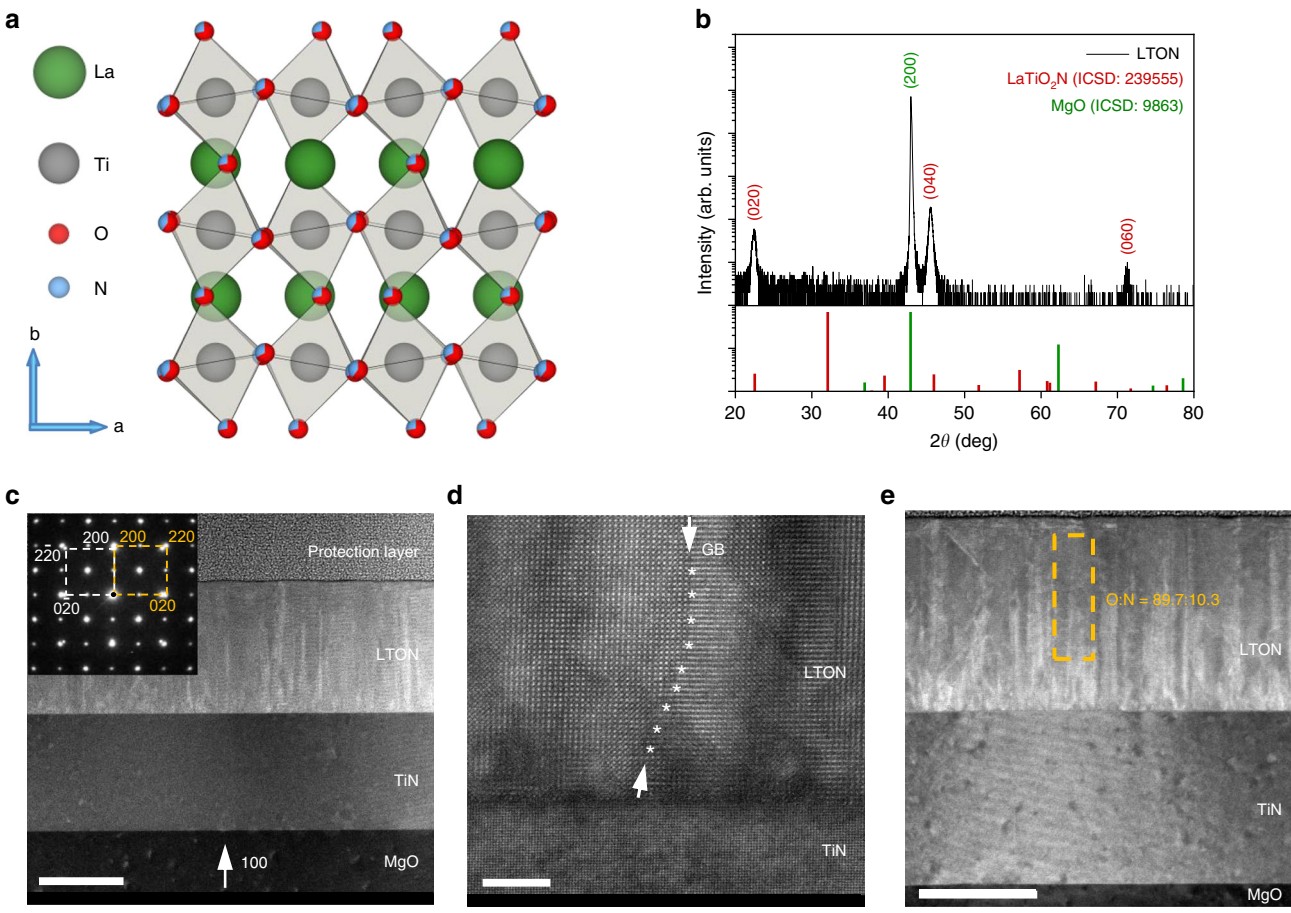

**Fig. 1 Structure and morphology characterisation of LTON. a** Crystal structure of LaTiO₂N. **b** XRD pattern of the epitaxial LTON film, the $\theta/2\theta$ is shown in black, with LaTiO₂N and MgO substrate reflexes shown below in red and green for reference. Reflections are labelled using the (hkl) notation. **c** TEM cross section of the LTON–TiN film grown on MgO. The scale is 100 nm. **d** High-angle annular dark-field (HAADF) image; a grain boundary (GB) is tracked with asterisks. The scale is 5 nm. **e** O:N ratio determined by the TEM/EELS analysis; the scale is 100 nm.

easily distinguish between the surface and the bulk of the material. With respect to powders, decreasing the nanoparticle radius size does increase the surface area-to-bulk ratio; however, any measurements would still represent an average of the contributions from the surface and the bulk. Literature pertaining to oxynitride thin films for water splitting remains fairly limited to date[20,23,24].

Some of the most prevailing surface-sensitive techniques currently available are not without their limitations. Secondary ion mass spectrometry (SIMS) for example involves the surface bombardment with a primary ion beam in an ultra-high vacuum. Therefore, the technique is locally destructive due to sputtering effects that can lead to modified surface chemistry and ion implantations. X-ray photoelectron spectroscopy (XPS) is another key technique for surface characterisations, though like SIMS it requires a high vacuum environment. These environments do not simulate natural operating conditions and can alter the sample itself, whilst also ruling out future in situ and operando measurements performed in water. Recent advances in Near Ambient Pressure XPS (NAP-XPS) have been made using dip and pull methods in water and offer promising opportunities for in situ measurements; however, one must consider changes in pH values, drying and salt formation. XRD is limited to long-range order and X-ray reflectivity (XRR) is not sensitive enough to distinguish between elements next to one another in the periodic table. Surface-enhanced Raman spectroscopy (SERS) and more specifically tip-enhanced Raman spectroscopy (TERS) is another

good possibility; however, the Raman effect is also very weak, and can require a long measurement time not suitable for operando measurements. The sensitive nano tips used can be prone to unwanted chemisorption and degradation during operation conditions.

Taking into consideration the type of information that can be obtained from the current existing techniques, their limitations and practicalities, we chose for this study to employ the complementary surface-sensitive techniques, NR and GIXAS, to probe the surface modifications of epitaxially grown LTON films (Fig. 2). The advantage of using NR and GIXAS is the fact that they are both in situ techniques, allowing the possibility to study reactions in the presence of water. Neutron reflectometry is a non-destructive[27] technique similar to XRR for measuring the depth profile of the density of thin films. The difference between using neutrons over X-rays is that the neutrons scatter from the atomic nuclei rather than from electrons. The neutron scattering power can vary greatly between neighbouring elements and isotopes of the same element; therefore, NR, unlike XRR, is sensitive to neighbouring light elements (H, C, O, N) and isotopic substitution[27–29] and can provide depth profiles with sub-Ångstrom resolution[30].

In NR, the neutron wave ($k_0$) hits the sample at a given angle. Part of the neutron wave is reflected ($k_f$) and part is refracted ($k_1$) (Fig. 2b). The scattering plane is orientated vertically ($q_z$) and the incident angle ($\omega$) can be by rotating the sample (Fig. 2a). Diffuse scattering has been omitted and only the specular reflected waves

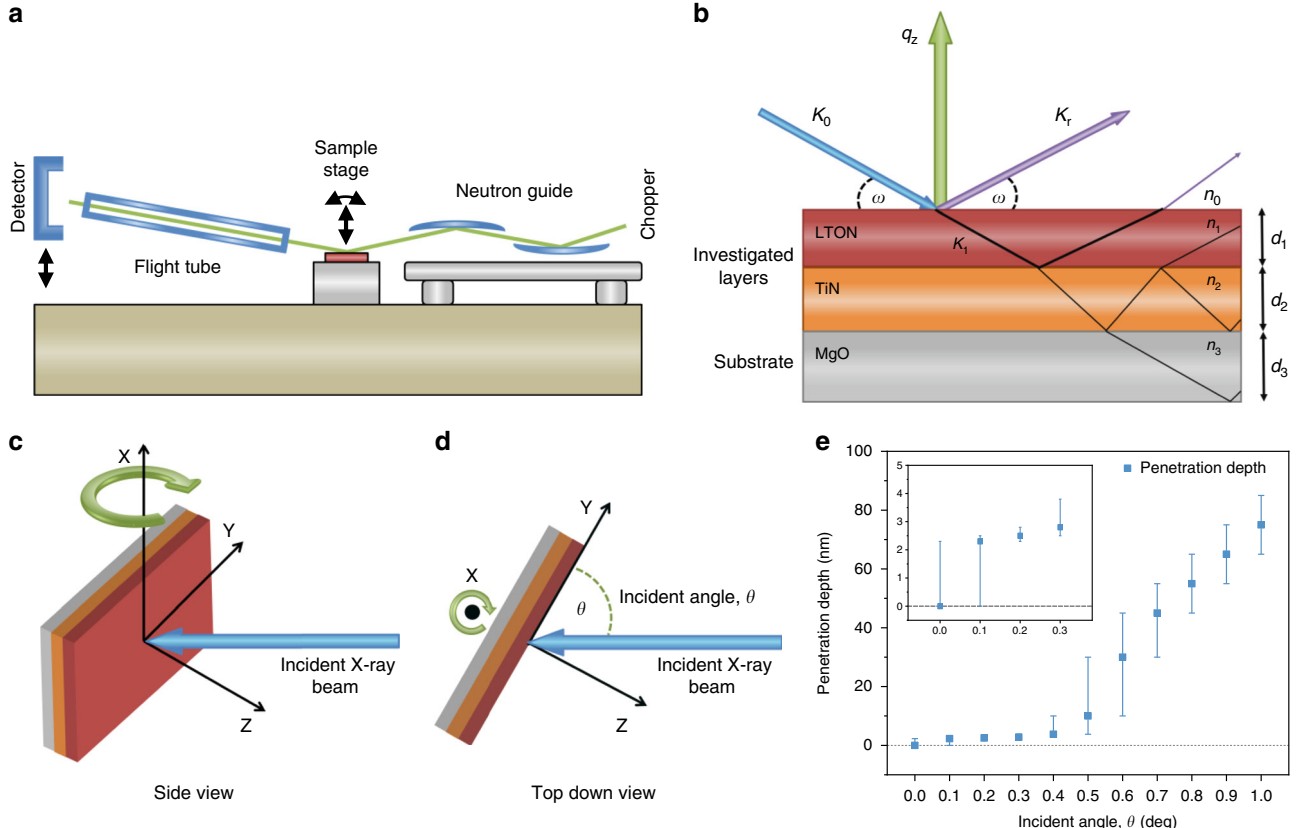

**Fig. 2 Experimental overview. a** Instrument description for AMOR time-of-flight neutron reflectometer at SINQ/PSI. **b** Neutron reflectometry measurement for multilayer thin films. **c**, **d** Surface-sensitive GIXAS measurement, side view and top down respectively. **e** Penetration depth of X-rays for LTON at the Ti K edge energy range as a function of incident angle, the error bars correspond to the penetration depth at a given incident angle with an error of ±0.1 degrees. Values taken from ref. [46].

are considered for easier visualisation. The depth profiles are obtained via specular NR by measuring the intensity of the reflectivity. From the reflectivity curves, scattering length density (SLD) profiles of the material are inferred.

XAS is an element-specific spectroscopic technique that allows the characterisation of the electronic structure of the absorbing atoms and of their local geometrical environment. To perform surface-sensitive measurements, grazing incident X-ray absorption spectroscopy (GIXAS) was performed on the LTON thin films, while powder reference samples were measured by XAS in transmission mode. Measurements were performed at grazing incident angles varying between 0 and 1° at intervals of 0.1° (Fig. 2c–e). This method enables us to distinguish between the surface and the bulk of our samples, a type of characterisation that is not possible with powder samples.

**Photocatalytic water splitting and photoelectrochemical characterisation.** The photocatalytic conversion of solar energy into hydrogen via the water-splitting process assisted by semi-conductor catalysts is a promising route to generate a clean, low cost and renewable energy source. In these semiconductor photocatalysts, electrons are excited from the valance band to the conduction band forming electron−hole pairs ($e^-$−$h^+$) under light irradiation (photons) with energy greater than or equal to that of the band gap (Fig. 3a). The photogenerated electrons are collected by the TiN seed layer and transferred to the Pt counter electrode where the excited electrons can directly reduce protons to form $H_2$. As for the photogenerated holes, they are consumed in the OER where they oxidise water to generate $O_2$ at the surface of the photocatalyst. Co-catalysts can also be used in conjunction

with the photocatalysts (not shown), which provide alternative active sites for the evolution of hydrogen and oxygen by acting as electron and hole scavengers for the photogenerated charges, respectively. Thus, co-catalysts can increase performance by reducing electron/hole charge carrier recombination which is a detrimental effect that suppresses the photocurrent.

The PEC characterisations of LTON were performed with a standard three-electrode configuration in 0.5 mol. NaOH aqueous electrolyte (pH = 13) as shown in Fig. 3b. A reversible hydrogen electrode sets the 0 V potential and an increasing voltage bias is applied between working and counter electrode. The electronic current between working and counter electrodes, also called the photocurrent, is proportional to the amount of $H_2$ and $O_2$ produced. Figure 3c shows the first three consecutive potentio-dynamic current−voltage scans for LTON. From the initial scan it is apparent that there is a decrease in the measurable photocurrent density with successive sweeps. By the third scan the photocurrent density is stable however, with a photocurrent density at 1.5 V vs. RHE ca. 20% less than the initial value. This degradation was previously ascribed to the surface oxidation of the epitaxial LTON film under operation conditions.

Measurements were performed under chopped illumination allowing the distinction between the dark current when there is no illumination of the sample, and the visible-light response that can be seen by the sharp increases in current when under illumination. Figure 3d shows the stabilised photocurrent density for the bare LTON sample compared to LTON decorated with $IrO_2$ nanoparticle co-catalyst deposited onto the surface, shown red and blue respectively. Figure 3e shows the degradation in photocurrent observed previously in Fig. 3c as a percentage from

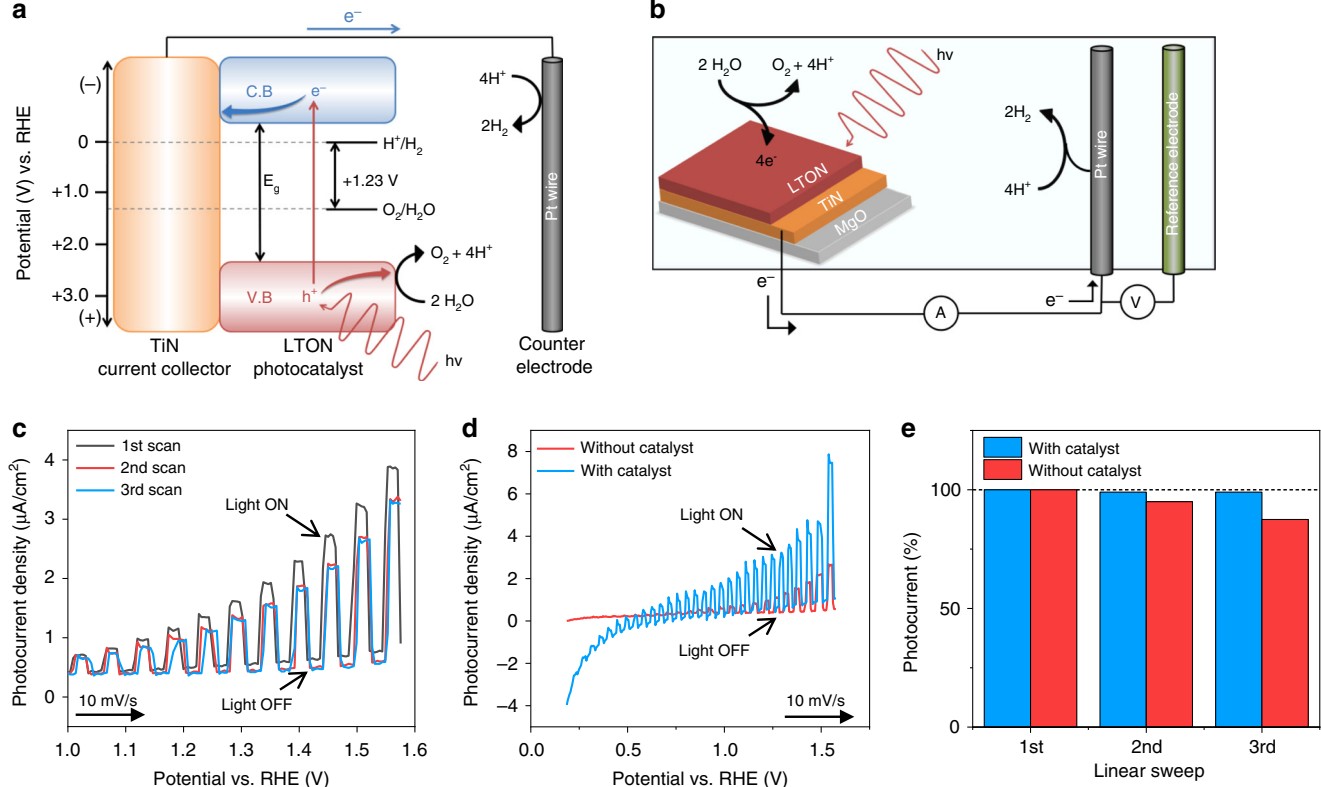

**Fig. 3 Photoelectrochemical water splitting. a** Energy diagram for the LTON semiconductor photocatalyst. **b** Photoelectrochemical three-electrode cell.
**c** Three consecutive potentiodynamic scans for LTON without catalyst. **d** Stabilised photocurrent for the bare LTON film and IrO₂ decorated LTON film shown in red and blue respectively. **e** The degradation from the initial photocurrent with successive potentiodynamic linear sweeps comparing the bare and IrO₂ decorated LTON films values taken at 1.5 V vs. RHE.

the initial measured photocurrent during the first linear sweep for the bare LTON film compared to the IrO₂ decorated film. It is apparent that the bare LTON film is more susceptible to the degradation process compared to the sample decorated with co-catalyst, where the degradation rate is decreased and the extent of the degradation is minimised. It is shown in Fig. 3d that in the presence of the co-catalyst, the photocurrent density is increased by a factor of ca. 2 at 1.5 V vs. RHE. Optimal co-catalyst deposition conditions and cheaper alternatives to IrO₂ are currently under study. The significance of the co-catalyst here is to compare the effect of the PEC water-splitting process on bare LTON thin films and those decorated with a well-known co-catalyst to compare performance and the extent of the surface modifications. To confirm and monitor any changes in, or the formation of a modified surface layer of the LTON films, NR measurements were performed on several LTON films ex situ comparing before and after PEC measurements.

**Neutron reflectometry**. We performed ex situ NR measurements on several LTON thin films; a visual description of the measurement has been included in Fig. 2. Couples of nominally identical samples were fabricated to compare NR results before and after PEC measurements. For one batch of samples, PEC characterisation was performed as described in the experimental section. For the second batch, the sample history was kept identical. It was exposed to the same electrolyte for the same time period and irradiated with the same light intensity. However, no external bias was applied to the film and no measureable photocurrent was observed, the reason for doing so was to try establishing which stimulus is the driving force for the observed surface degradation. The resulting reflectivity plots for the before

and after comparisons of the two sets of films are shown in Fig. 4a–b. It can be seen in Fig. 4a that for the sample that did not evolve any oxygen there is minimal observable changes in the reflectivity curves when comparing the initial as-grown film and after exposing to the electrolyte and light irradiation. At higher Q (scattering vector) values there are small deviations but they are smaller than the resolution and degree of error as the signal tends to wash out at high Q values. The difference spectrum has been provided for easier visualisation. When comparing a sample that was exposed to an external bias and evolved oxygen under OER conditions, significant changes in the reflectivity curve can be observed when comparing before and after PEC measurements (Fig. 4b).

These differences arise due to changes in the density of the film during operation. The extent of the changes can be described by modelling of the reflectivity curves. Figure 4c shows the experimental data (solid lines) from Fig. 4b with the corresponding model fit (dashed lines) which has been offset in the y-axis for easier visualisation. From the fit, a 1D SLD profile is inferred and is portrayed in Fig. 4d. Comparing before and after PEC the TiN layer remains unchanged, contrast differences in the SLD profile at each TiN interface arise due to the slight diffusion of oxygen to the TiN layer from the substrate and from the LTON film during growth which has also been evidenced previously by TEM and electron energy loss spectroscopy (EELS) (Supplementary Fig. 2). Comparing before and after PEC for the LTON film, a minor difference is apparent at the TiN/LTON interface inferred by an almost negligible O:N ratio change but the largest changes in SLD profile occur at and just below the surface, as seen in Fig. 4d and the inset. The model suggests slight nitrogen content loss in the film below the surface layers, but at the surface within the first

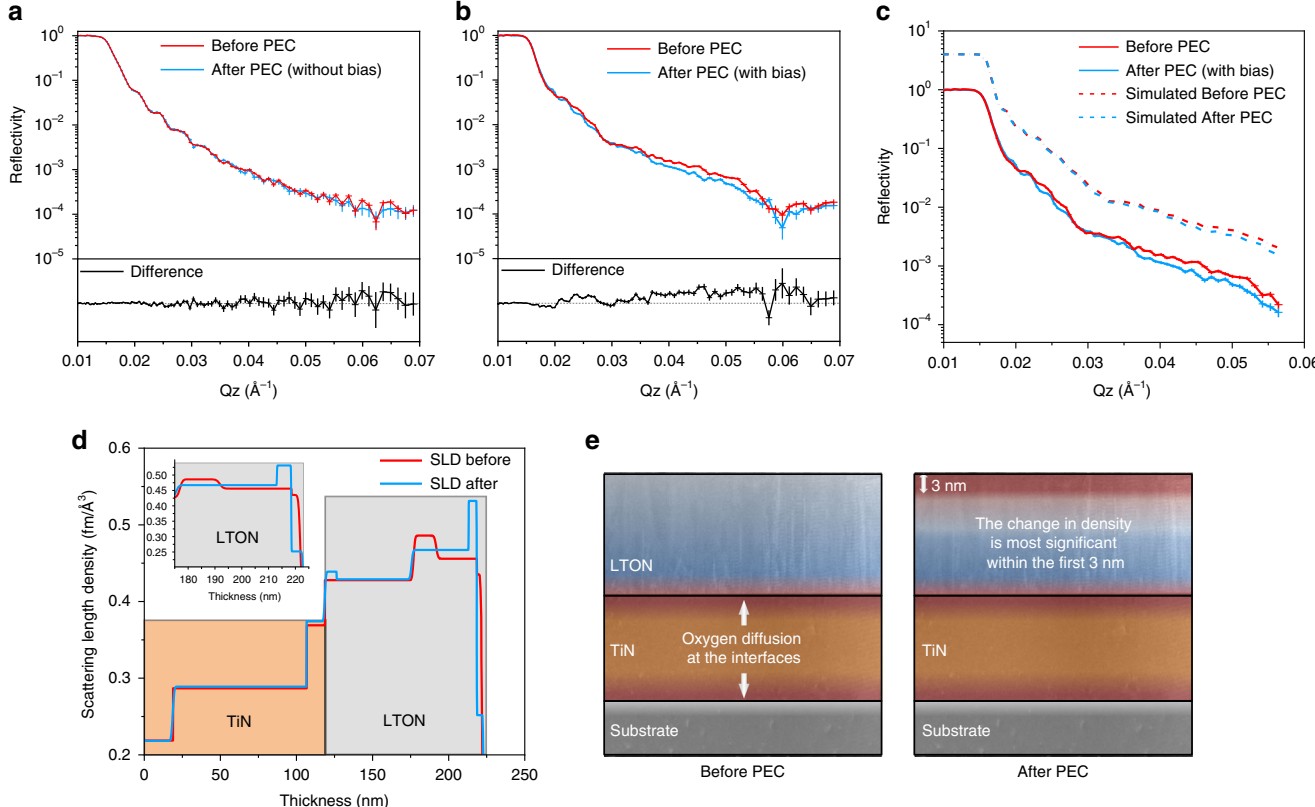

**Fig. 4 Neutron reflectometry data for LTON thin films. a** Reflectivity plot for an LTON film exposed to electrolyte and light intensity, yet no external bias was applied to the sample and showed no photoelectrochemical performance. **b** Reflectivity curve for LTON thin film comparing before and after PEC measurements with applied bias. **c** Experimental and corresponding model fit for data shown in (**b**), shown by the solid and dashed lines respectively. **d** The inferred scattering length density profile as a function of depth from the corresponding fit. The inset shows the magnified region at and near to the surface of LTON. **e** Depth profile of the samples before and after PEC, where the largest change in the SLD corresponds to a more oxidised surface layer of 3 nm.

3 nm there is a greater degree of nitrogen loss and a slight increase in the oxygen content which would generate a small degree of vacancy formation in the film after PEC measurements. However, the quantitative analysis of the stoichiometric changes resulting in the scattering length density profile changes involve some degree of uncertainty and require additional complementary techniques to reference.

Qualitatively, the differences in the scattering length density profile confirm that a physiochemical change in the film has taken place, most significantly at and near the surface where the model suggests slight stoichiometric changes. Since no difference was observed in the film where no external bias was applied, we could conclude that the density changes observed for the oxynitride are of a direct consequence of the applied external bias and the water-splitting process. The electrolyte and incident light alone has minimal/negligible effect on the LTON sample. Although NR is a powerful technique with various applicational benefits, relying solely on NR is not sufficient to elucidate the degree of changes in the LTON films.

**X-ray absorption spectroscopy.** For this study, we performed ex situ GIXAS measurements on several LTON thin films before and after PEC, to probe the electronic structure of the La and Ti, A and B cations and their local environments. Figure 5 shows the X-ray absorption near edge spectroscopy (XANES) spectra for LTON measured at the lanthanum $L_3$ edge and Ti K edge including several powder references for both elements.

Figure 5a compares several La-containing references. There is no significant shift seen in the energy position of the absorption edge peak at ca. 5.49 keV for all three samples. This suggests that

lanthanum exists in the commonly observed 3+ oxidation state for the prepared powders of the LaTi oxide and oxynitride materials. There are, however, significant differences in the intensities of the peaks for the three samples, which are attributed to the different and perhaps non-uniform particle sizes. Different size particles possess different surface areas and as a result can exhibit partial oxidation reflected in the changes of the peak intensities. Figure 5b shows the comparison of the bare LTON film with the $IrO_2$ decorated film measured in the bulk of the sample.

It can be seen that after PEC the La cations in the bulk of the bare sample undergo both an increase in intensity and a reductive shift to lower energy. For the La cations situated at the surface however, it can be seen that there is an oxidative shift to higher energy after PEC (Fig. 5c). It is noteworthy that when the LTON films are decorated with the $IrO_2$ co-catalyst, the shifts in the edge positions and intensity changes are prevented.

It has previously been shown that, looking at the second derivative plots of the La XANES and peak fitting, analysis looking at peak positions and FWHMs are related with the degree of disorder and coordination environment surrounding La[31,32]. The same analysis was carried out on the La $L_3$ XANES data presented here but it did not show any significant difference comparing between before and after PEC. The analysis has been included in Supplementary Fig. 3.

The biggest differences in the XANES spectra are the centroid shifts which suggests that the La A cation sites are much more active in the water-splitting process than previously believed. This was not obvious or expected for the A site and especially for La. The B cation would correctly be assumed to be more catalytically active than La, but it has been previously shown that the surfaces

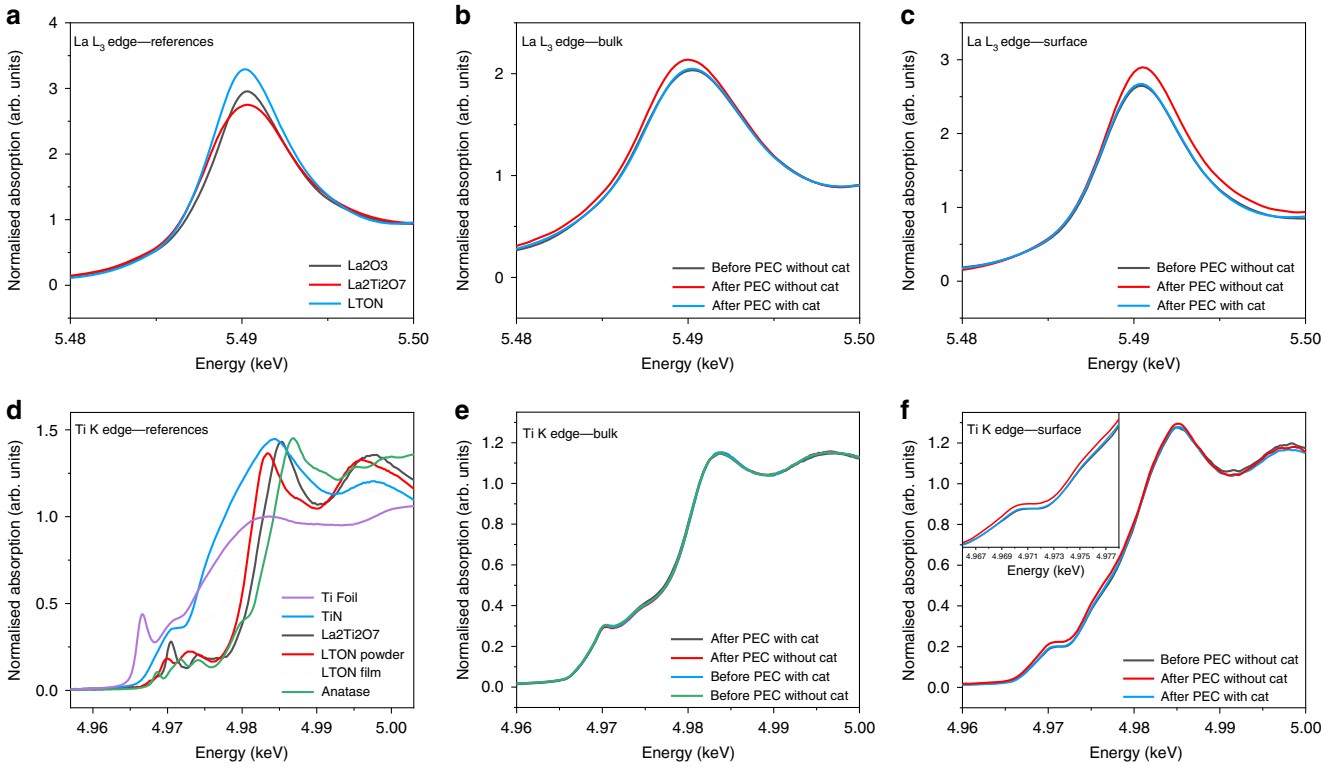

**Fig. 5 XANES spectra for LTON measured at the La L₃ edge and Ti K edge with references. a** La-containing references. **b** LTON bulk comparing La before and after PEC. **c** LTON surface comparing La before and after PEC. **d** Ti-containing references. **e** LTON bulk comparing Ti before and after PEC. **f** LTON surface comparing Ti before and after PEC.

of these oxides and oxynitrides tend to show AO (La-O) termination rather than BO₂ (TiO₂) surface terminations[24,33–35]. The La cations are much larger than Ti (1.03−1.16 Å and 0.61 −0.67 Å respectively) and so the surface would consist primarily of La-O atoms where the Ti B cations sink down slightly into the sub-surface layers, which could explain why the A site shows evidence to be the catalytically active site for the OER.

Traditionally, it is assumed that the valence state of lanthanum exists solely in the nominal 3+ integer value. However, there have been theoretical and experimental studies that suggest that this is not entirely true. It has been shown that La can exhibit a partially occupied d-orbital (5d1) with an atomic charge between the nominal valence charges +2 and +3[36–38]. This partial covalency between La-O would explain the observed catalytic behaviour of the A cation and the changes seen in the XANES spectra.

The reductive shift in bulk of the samples may be driven by the slight loss of nitrogen/oxygen and generation of vacancies changing the partial covalency surrounding La. It is possible and worth noting that overall nitrogen contents could be retained, where nitrogen leaves the lattice but remains in interstitial sites. At the surface the La oxidative shift in centroid position could be due to the adsorption of *O, *OH, *OOH intermediates under OER conditions where adjacent O decouples to release molecular O₂.

Using a co-catalyst conserves the valence state of lanthanum after PEC measurements as seen by the edge positions remaining static and not shifting as seen for the bare sample by filling the role of the active site rather than the A cations. The same measurements were performed at the titanium K edge to explore the B cation site in the LTON films and are portrayed in Fig. 5d–f. Figure 5d shows a number of Ti-containing references for comparison to the LTON films. As expected the Ti valence states in the perovskite powder samples and the LTON film are a mixture of 3+ and 4+. This can be seen by edge positions and

peak intensities that are situated in-between those of Ti³⁺ in titanium nitride and Ti⁴⁺ in anatase. The comparison between the bare and co-catalyst-loaded LTON films measured in the bulk before and after PEC characterisation (Fig. 5e) conveys that the water-splitting process has no effect on the valence state of the titanium cations in the bulk of the samples. Figure 5f shows the comparison of the LTON films measured at the surface with and without the co-catalyst before and after PEC measurements. It is evident from the spectra that there is no shift in the edge position, meaning that the valence state of Ti has not changed as a consequence of the water-splitting process.

There are, however, slight changes in the intensity of the pre-edge region (4.96−4.98 keV) as a consequence of the PEC measurements. Since pre-edge features are generally related to symmetry and local coordination environments, it is suggested that as a consequence of the OER there is an increase in the disorder of the Ti octahedra located at the surface of the LTON film. This type of information and data are not obtainable with some of the previously described surface-sensitive techniques. Measurements were performed at various incident angles as previously discussed and the observed changes in the pre-edge for Ti were limited to within the first 3 nm of the film. The nature of the changes in the pre-edge features will be discussed in the following section but first when comparing the bare LTON sample with the sample decorated with the co-catalyst it is noteworthy that when the co-catalyst is present the pre-edge features changes previously seen are subdued in the presence of IrO₂.

**Pre-edge features**. In order to understand the extent of the changes in the pre-edge features observed at the surface in the Ti K edge spectrum after PEC measurements as seen in Fig. 5f, the

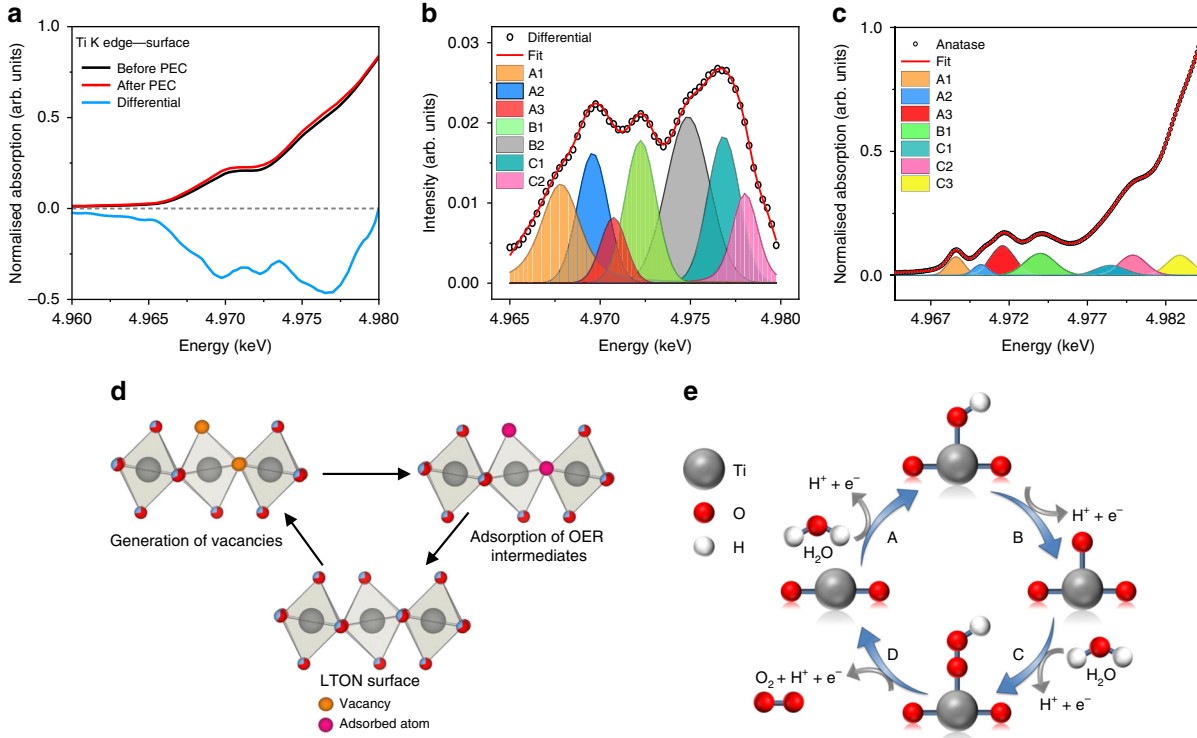

**Fig. 6 LTON Ti K edge pre-edge differential and peak fit. a** The bare LTON film surface before and after PEC and the differential spectra shown in black, red and blue respectively. **b** Differential spectra and corresponding peak fitting analysis. **c** Peak fitting analysis for measured $TiO_2$ anatase as a reference for peak assignments. **d** Ti(O/N)$_6$ octahedra and the possible axial and equatorial positions for vacancy generation and adsorption sites. **e** Schematic representation of the OER mechanism.

differential spectrum for the interested region has been produced and displayed with the original data as shown in Fig. 6a. The differential and the corresponding peak analysis fit can also be seen in Fig. 6b.

The pre-edge peak intensities arise due to disorder, lattice defects[39,40] and a lowering of symmetry, along with orbital mixing. Therefore, the intensity of these peaks increases considerably when moving away from a perfect octahedral to tetrahedral Ti environments. The pre-edge peaks in Fig. 6b correspond to the 1s > 3d, 1s > 4s and 1s > 4p transitions of the core 1s electron to the unoccupied molecular orbitals of the Ti(O/N)$_6$ octahedra. Their assignments are summarised in Supplementary Table 2 and discussed in more detail below. Each peak corresponds to a certain transition and is in agreement for peak fitting analysis performed on experimentally obtained data for $TiO_2$ (Fig. 6c). These transitions have also previously been discussed in literature[39]. The exception is peak B2 that contributes to the differential fitting for the LTON thin film but has no evidence of any contribution to $TiO_2$ pre-edge analysis.

**Peak A1**. The peak denoted as A1 has previously been assigned to a 1s-3d transition usually forbidden in perfect octahedra but due to disorder and a lowering of symmetry, along with 4p-3d orbital mixing with neighbouring Ti atoms, the peak is weakly present for LTON and in $TiO_2$. The intensity of this peak increases considerably as you move away from an octahedral to tetrahedral Ti environments. After PEC measurements the contribution of this peak has increased suggesting an increase in disorder of the Ti(O/N)$_6$ octahedra of the Ti atoms located at the surface.

**Peak A2**. The peak denoted as A2 has previously been assigned to a 1s-3d quadrupole transition for $TiO_2$ and is related to lattice defects. Perhaps more specifically for LTON with D$_{2h}$ symmetry,

the A2 peak can be assigned as the 1s-3d (b$_{3g}$ d$_{yz}$) which is primarily quadrupolar in character. After PEC measurements, the intensity of this peak increases. Since it has been previously assigned to defects in $TiO_2$, it therefore suggests that after the water-splitting process the local geometry surrounding the surface and near-surface Ti cations become a little more disordered. Increasing disorder lowers the octahedral D$_{2h}$ symmetry even further, introducing more dipolar character to the A2 peak by p-d orbital mixing. This could account for the observed intensity increase of this peak as a consequence of the water-splitting process.

The reason for increase in disorder could also be explained by nitrogen loss from the lattice and the formation of nitrogen/oxygen vacancies. This would lead to pentacoordinated Ti cations, lowering the symmetry and increasing p-d orbital mixing. It has been previously shown by ab initio FDM calculations for $TiO_2$[41] that oxygen vacancies can lead to strong enhancements in the A2 peak region due to the lowering of symmetry and a spectral shift of the A1 peak for pentacoordinated Ti atoms with oxygen vacancies. This may explain why this peak is most significant for the LTON thin films but minimal in powdered anatase.

**Peak B1**. Peak B1 is attributed to the non-local 1s to 4p-4s hybridised states with the second Ti neighbour. From the peak fitting it suggests that the B1 peak also increases in intensity after PEC which could be accounted for with the increase in disorder around the local absorbing Ti atom altering the degree of p-d mixing with neighbouring Ti atoms.

**Peak B2**. The B2 peak that is present only in the LTON film can be explained by the existence of oxygen/nitrogen vacancies. This is supported also by the ab initio *FDM* calculations previously

described under section "Peak A2". According to the calculations, the introduction of axial and equatorial oxygen vacancies leads to large peak intensities for $TiO_2$ situated at ca. 2.5−3 eV higher in energy than B1. For LTON the difference between B1 and B2 is also separated by a similar energy difference (Supplementary Table 2). Since this contribution is suggested to be due to oxygen vacancies, the increase in the intensity of this peak after PEC measurements seems to suggest that the OER creates oxygen and/ or nitrogen vacancies driven by the loss of nitrogen at the surface of the oxynitride films. This would agree with the NR model that the surface is comprised with an increased oxygen content.

**Peak C1**. The C1 peak situated at ca. 4977.2 eV also has a large change in amplitude after PEC measurements. This peak is characterised for $TiO_2$ as the 1s to 4p ($B_{1u}$) transition, which is dipolar in character. The antibonding $B_{1u}$ molecular orbital is comprised of the Ti $4P_z$ orbitals and N/O $2P_z$ orbitals. The large increase for this transition may be explained by the loss of N/O and the creation of vacancies at the apical position of the Ti octahedra where the surface terminates and adsorption of intermediates during the OER takes place.

The contributions of these peaks seem large due to the fitting of the difference spectra, the overall disorder of the Ti environment as a consequence of the water-splitting process are quite subtle when looking at the XANES data. However, we would like to highlight the fact that the edge position does not change, yet XAS is a suitable technique to detect this short-range structural information not seen by XRD or XPS. The change in stoichiometry and the increase in disorder of the environment surrounding Ti would induce changes in the degree of local strain of the films. These physiochemical changes can account for the reduced OER activity by increasing the binding energies of the *O, *OH and *OOH intermediates and increasing the OER overpotentials.

## Discussion

We have applied complementary NR and surface-sensitive grazing incidence absorption spectroscopy techniques to probe the detrimental surface modifications associated with oxynitrides that occur during operation. We show that thin films are ideal model systems for surface-sensitive studies where one can more easily distinguish the surface signals from the bulk contributions. The physiochemical modification of the oxynitride surface was limited to the first 3 nm of the bare thin film. The addition of the $IrO_2$ co-catalyst not only increases the performances of the oxynitride sample by a factor of two but also protects the surface by preventing the observed surface modification as a direct consequence of the water-splitting process. The surface modification involves the oxidation of the La A cations and a disordering of the local environment surrounding the Ti B cations which are thought to be detrimental and the reason for the reduced performance observed in the oxynitrides, whereas La undergoes a reduction in the bulk and Ti remains unaffected as a consequence of the water-splitting process. These findings were unexpected due to the fact that we have evidence that the A site plays a significant role in the OER and La does not exist solely in the nominal +3 valence state. The reports in literature tend to focus more on the role of the B cations, with many new materials involving doping and/or partial substitutions at the B cation sites with respect to activity.

To the best of our knowledge, this is the first depiction of what is occurring at the surface and the extent of the changes with respect to each cation as a function of depth during photocatalytic water splitting. Studies into the optimisation of co-catalyst deposition conditions, as well as surface-sensitive GIXAS, operando water-splitting measurements with a custom built in situ

cell reactor are ongoing, including operando extended X-ray absorption fine structure measurements for increased information regarding the evolution and the extent of the structural changes at the surface.

## Methods

**Thin film deposition**. The LTON films were grown using a modified pulsed laser deposition (PLD) method called PRCLA previously described here[12] using a KrF excimer laser (Lambda Physik LPX 300, 30 ns pulse width, $\lambda$ = 248 nm) to ablate a target of $La_2Ti_2O_7$ fabricated in our laboratory. The target to substrate distance was set at 50 mm. A laser fluence of 3 J/cm² with a laser repetition rate of 10 Hz was applied. Commercially available (001)-oriented MgO was used as a substrate ($10 \times 10 \times 1$ mm). Platinum paste was applied between the substrates and heating stage to provide good thermal conductivity. The substrate temperature was set at 750 °C measured via a pyrometer. $N_2$ partial pressure of $8.0 \times 10^{-4}$ mbar was set via a gas inlet line to the vacuum chamber. $NH_3$ gas jets were injected through the piezo-controlled nozzle valve of the gas pulse. The opening time of the nozzle valve was set at 400 μs and the delay between the valve closing and the laser pulse was set at 30 μs.

The titanium nitride current collector layer was grown by conventional PLD using a commercially available TiN target under vacuum with a base pressure of ca. $5 \times 10^{-6}$ mbar. The substrate temperature was set at 750 °C. A laser fluence of about 3 J/cm² with a laser repetition rate of 10 Hz was applied.

In this work we used three independent sets of samples that were characterised in three dedicated measurement campaigns at the synchrotron light source and two at the neutron source. The best data were acquired with the third set of samples after optimisation of the experimental set-ups.

**Photoelectrochemical characterisation**. Photoelectrochemical measurements were performed using a three-electrode configuration. The working and counter electrode were the LTON thin films and Pt wire respectively. A KCl saturated Ag/ AgCl electrode was used as the reference. An aqueous solution of 0.5 mol. NaOH (pH = 13) was used as an electrolyte. For the electrical contact of the LTON films, on one side of the films, a small section of LTON was carefully removed to expose the TiN current collector to apply the electrical contact to the potentiostat (Sola-tron 1286). The electrically connected area was then insulated with epoxy and the sample then immersed into the electrolyte. The samples were illuminated with a 150 W Xe arc lamp (Newport 66477) with an AM 1.5 G filter with an output intensity of 100 mW/cm². To measure the dark and light current, a chopper was used to intermittently block the irradiation of the sample. The potentiodynamic and potentiostatic measurements were performed at a scan rate of 10 mv/s. $IrO_2$ nanoparticles were deposited by immersion in an $IrO_2$ colloidal solution for 30 min. The $IrO_2$ solution was prepared by hydrolysis of $Na_2IrCl6·6H_2O$ (0.010 g) dissolved in $H_2O$ (50 mL), and the pH was adjusted to 11−12 with NaOH (1 M). The solution was heated at 80 °C for 30 min and cooled to room temperature by immersion in an ice water bath. The pH of the cooled solution was adjusted to 10 with $HNO_3$. Subsequent heating at 80 °C for 30 min resulted in a blue solution containing colloidal $IrO_2$.

**Chemical composition**. Metal ratios, oxygen content and film thickness were determined by Rutherford backscattering (RBS). Measurements were performed using a 2 MeV He beam and a silicon PIN diode detector. Data were analysed using RUMP[42]. Nitrogen-to-oxygen ratios were determined by Elastic Recoil Detection Analysis (ERDA) using a 13 MeV $^{127}I$ beam, time-of-flight spectrometer and gas ionisation detector.

**Crystalline properties**. XRD measurements were performed using a Bruker-Siemens d500 X-ray diffractometer with characteristic Cu $K_\alpha$ radiation 0.154 nm. Unlocked coupled $\theta$-$2\theta$ scans were performed to determine the out-of-plane orientations of the films. Grazing incidence detector scans ($\theta$ = 1) were also performed to determine whether a film crystalline quality is epitaxial, textured or polycrystalline.

**Transmission electron microscopy**. Transmission electron microscope specimens were prepared by focused ion beam lift-out technique using a Thermofisher Sci-entific Helios UC instrument operated at 30 kV at the beginning of preparation followed by 5 and 2 kV for the final cleaning. The HAADF and EELS images were collected with a Thermofisher Scientific (former FEI) Titan 80-300 environmental TEM operated at 300 kV, equipped with a Gatan Imaging Filter Quantum 965 ER.

**Neutron reflectometry**. Neutron reflectivity (NR) measurements were performed at the reflectometer AMOR at SINQ, Paul Scherrer Institut in Switzerland[43]. There the neuron beam is focused to the sample by elliptic reflectors leading to a divergence of 1.6°. The angular resolution is realised via the position sensitive detector. The intensity of the reflected neutron beam from the sample is measured as a function of the scattering angle and the time-of-flight from the chopper to the

detector. The wavelength resolution was 5% and measurements were performed over a Q-range (1/Å) ranging between 0.005 and 0.1.

**X-ray absorption spectroscopy**. XAS measurements were performed at the SuperXAS beamline at the SLS, Paul Scherrer Institut in Switzerland. The energy resolution was optimised using (111)-Si monochromator crystals. Energy calibration was performed using a thin Ti reference foil. A focused beam was used with a spot size of $100 \times 100$ μm$^2$. The samples were mounted in grazing incidence to the incoming X-ray beam, with an incident angle between 0° and 1°. XAS spectra were obtained in fluorescence yield using a five-element SDD detector situated at 90° to detect fluorescence. Data analysis was performed using the Athena and Larch software packages[44,45].

## Data availability

The data used to support this study are available from the corresponding authors upon reasonable requests.

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

## Acknowledgements

The authors would like to thank the Paul Scherrer Institute, and the financial support by the Deutsche Forschungsgemeinschaft, grant number CRC 1073 (Projects Z02) is highly appreciated.

## Author contributions

Conceptualisation, C.L., D.P., J.S., and M.N.; Investigation, C.L., M.N., V.R., M.D. and J.S.; Formal analysis, C.L., M.N., M.D., J.S.; Writing—original draft, C.L. and D.P.; Writing—review & editing, C.L., T.J.S., D.P and T.L.; Funding acquisition, D.P., J.S., M.N., T.J.S.; Supervision, D.P. and T.L.

## Competing interests

The authors declare no competing interests.
