## [Peer Review File · Nature Communications]

Reviewers' comments:

Reviewer #1 (Remarks to the Author):

With the described surface sensitive methods (NR and GIXAS) to study the physicochemical processes occurring at the SC/liquid interface, it is possible to reveal the alteration of the surface during the PEC reaction and also possible to identify the mechanism of photoelectrocatalyst degradation. Thus, I think that this study will attract the interest of the scientific community. However, there are two key steps which in my opinion would make even a bigger impact:

- 1.) To show data on in-situ measurements
- 2.) To demonstrate versatility of these combined techniques on other SCs (I understand, that perovskite-type oxynitrides are promising materials due to their optimal bandgap and band edge positions, however, to put it mildly, their potential is not fully exploited. Also, it is not elegant to show photovoltammograms with 1-2 microA/cm-2 photocurrents which is then changing to 5-6 after adding a co-catalyst to the system).

Overall, the authors have addressed all critical comments and the manuscript is suitable for publication in Nature Communications

Reviewer #3 (Remarks to the Author):

The paper of C. Lawley emphasizes, for the first time, by means of neutron reflectometry and grazing incidence X-ray diffraction spectroscopy, an oxidation of the La (A) cations at the surface of perovskite oxynitride LaTiO₂N thin films undergoing water-splitting process, associated to a local disorder of the Ti (B) cations. Such processes are cancelled out when films are decorated with a co-catalyst, thus leading to a better photoelectrochemical response.

The study presented here is of quality and the article is well written. Nevertheless, despite novel and high quality results, I find that these are not at the level required for the journal Nature Communications and, so, I recommend to reject the article.

Three main remarks:

- As the authors write in their response to Reviewers, the signals are tiny. All their hypothesis and model are based on these tiny signals both in neutron reflectometry and grazing incidence X-ray diffraction. So the question arises on the reproducibility of the results. To the question n°2 of Reviewer 2, the authors respond that "we would not think to measure just one sample once", but no details are given on the exact number of LTON films that were measured.
- This leads to my second remark: the authors clearly evidenced that the epitaxial LTON films are no longer stoichiometric with regard to the nitrogen content (i.e. O/N = 10.3 – 10.7 and formulation La_{1.02}Ti_{0.98}O_{3.0}N_{0.28} in Supplementary Table 1). I wonder how this composition, surely associated with a defected perovskite structure, can itself be responsible to the oxidation of the La cations during the water-splitting process. Do stoichiometric LaTiO₂N films, even polycrystalline, respond as their epitaxial counterparts? This specific study would have been of interest. The additional remark of Reviewer 2 is then of great interest: do the cell parameters emphasize a (strong, weak) distortion of the perovskite ABO₂N cell and how does this influence the behavior of A and B cations during the photoelectrochemical test?
- As a third remark, I find that the very interesting parts, that is scientific detailed discussion of the results, are located in ... the Supplementary part. Readers get the impression that the results are presented in the manuscript as accepted, without detailed arguments. On the contrary, experimental results are well argued in Supplementary section (see for example "titanium K pre edge peak assignments"), but we should read the article without the help of the Supplementary add-ons.

So, as a conclusion, I find this article very interesting and innovative but the article should be rewrite

deeply saying clearly on how many samples the analysis has been undertaken, what is the structure of the under-stoichiometric LTON films and if it influences the photoresponse and associated processes on La and Ti cations. A robust and detailed argumentation has to be put inside the article.

Dear Dr Pergolesi,

Your manuscript entitled "Oxynitride solar water splitting photocatalysts: examining the surface evolution of LaTiO_xN_y " has now been seen by 2 referees. We note that one of the previous referees was unable to assess the work again, although we have found an additional referee to examine the work. You will see from their comments below that while they find your work of high interest, some important points are raised. Both referees have important suggestions we believe will improve the manuscript. For instance, we believe Reviewer #3's concerns regarding the reproducibility and the structure plus influence of the sub-stoichiometric films, are particularly important to address. We are interested in the possibility of publishing your study in Nature Communications, but would like to consider your response to these concerns in the form of a revised manuscript before we make a final decision on publication.

We therefore invite you to revise and resubmit your manuscript, taking into account all the points raised. Please highlight all changes in the manuscript text file.

We are committed to providing a fair and constructive peer-review process. Do not hesitate to contact us if you wish to discuss the revision in more detail or if there are specific requests from the reviewers that you believe are technically impossible or unlikely to yield a meaningful outcome.

Reviewers' comments:

Reviewer #1 (Remarks to the Author):

With the described surface sensitive methods (NR and GIXAS) to study the physicochemical processes occurring at the SC/liquid interface, it is possible to reveal the alteration of the surface during the PEC reaction and also possible to identify the mechanism of photoelectrocatalyst degradation. Thus, I think that this study will attract the interest of the scientific community. However, there are two key steps which in my opinion would make even a bigger impact:

- 1.) To show data on in-situ measurements
- 2.) To demonstrate versatility of these combined techniques on other SCs (I understand, that perovskite-type oxynitrides are promising materials due to their optimal bandgap and band edge positions, however, to put it mildly, their potential is not fully exploited. Also, it is not elegant to show photovoltammograms with 1-2 $\mu\text{A}/\text{cm}^2$ photocurrents which is then changing to 5-6 after adding a co-catalyst to the system).

Overall, the authors have addressed all critical comments and the manuscript is suitable for publication in Nature Communications

Reviewer #3 (Remarks to the Author):

The paper of C. Lawley emphasizes, for the first time, by means of neutron reflectometry and grazing incidence X-ray diffraction spectroscopy, an oxidation of the La (A) cations at the surface of perovskite oxynitride LaTiO_2N thin films undergoing water-splitting process, associated to a local

disorder of the Ti (B) cations. Such processes are cancelled out when films are decorated with a co-catalyst, thus leading to a better photoelectrochemical response.

The study presented here is of quality and the article is well written. Nevertheless, despite novel and high quality results, I find that these are not at the level required for the journal Nature Communications and, so, I recommend to reject the article.

Three main remarks:

- As the authors write in their response to Reviewers, the signals are tiny. All their hypothesis and model are based on these tiny signals both in neutron reflectometry and grazing incidence X-ray diffraction. So the question arises on the reproducibility of the results. To the question n°2 of Reviewer 2, the authors respond that “we would not think to measure just one sample once”, but no details are given on the exact number of LTON films that were measured.

- This leads to my second remark: the authors clearly evidenced that the epitaxial LTON films are no longer stoichiometric with regard to the nitrogen content (i.e. O/N = 10.3 – 10.7 and formulation $\text{La}_{1.02}\text{Ti}_{0.98}\text{O}_{3.0}\text{N}_{0.28}$ in Supplementary Table 1). I wonder how this composition, surely associated with a defected perovskite structure, can itself be responsible to the oxidation of the La cations during the water-splitting process. Do stoichiometric LaTiO_2N films, even polycrystalline, respond as their epitaxial counterparts? This specific study would have been of interest. The additional remark of Reviewer 2 is then of great interest: do the cell parameters emphasize a (strong, weak) distortion of the perovskite ABO_2N cell and how does this influence the behavior of A and B cations during the photoelectrochemical test?

- As a third remark, I find that the very interesting parts, that is scientific detailed discussion of the results, are located in ... the Supplementary part. Readers get the impression that the results are presented in the manuscript as accepted, without detailed arguments. On the contrary, experimental results are well argued in Supplementary section (see for example “titanium K pre edge peak assignments”), but we should read the article without the help of the Supplementary add-ons.

So, as a conclusion, I find this article very interesting and innovative but the article should be rewrite deeply saying clearly on how many samples the analysis has been undertaken, what is the structure of the under-stoichiometric LTON films and if it influences the photoresponse and associated processes on La and Ti cations. A robust and detailed argumentation has to be put inside the article.

Authors' response to the reviewer's comments:

Dear Dr. Weingarten,

Thank you for taking your time and consideration over our manuscript. We would also like to extend our thanks to all reviewers involved in the process so far for their fair and constructive comments. Please see below our additional responses to the reviewers comments on a point-by-point basis.

Reviewer 1 point 1:

We agree with reviewer 1 for the appeal of in situ and/or operando measurements. We are currently working on operando characterisations, thanks to the information and experience gained from the work in this manuscript and are extending this study to more materials. We kindly ask the reviewer to take into consideration also that this experimental approach would not in any case be appropriate for LTON. This is indeed one of the more interesting material for solar water splitting, but the energy for the Ti K edge is too low to measure in water unfortunately. Therefore, we are extending the study to oxynitrides having Ta and Nb in the B site (such as LaTaOxNy, SrTaOxNy, BaTaOxNy or CaNbOxNy), and to different oxide based semiconductors. However, this will be a separate study, which we hope to finalise in the near future.

Reviewer 1 point 2:

The photocurrents are low as highlighted by the reviewer when compared to powder based samples. However, as discussed in the manuscript there is a large difference in the active surface area between thin film and powder samples, ca. 50-100 times smaller for the thin films. Therefore, a few microAmps measured with a thin film with a co-catalyst corresponds to a few hundred microAmps for a powder sample. Moreover, in a thin film the N content can also be lower than for powder, making the sample less responsive to visible light.

The low photocurrent values are indeed in-line with what we expected and with what is reported in literature for similar samples. The thin films are certainly not made for performance, but this is the only sample design allowing grazing angle spectroscopy to achieve surface sensitivity.

We are currently working on performance enhancing strategies for the optimisation of the cocatalyst deposition conditions as well as exploring cheaper alternatives (Ni, Co, Fe based catalysts) both for powders and thin films for fundamental studies.

Reviewer 3 point 1:

Indeed this work is interested in the small changes that occur as a consequence of the applied bias and water splitting process. Concerning the X-ray spectroscopy data, the extent of the shift in the energy position of the absorption edges, as well as the small changes of the pre-edge features, are in the same range of which is typically portrayed in literature in almost all research fields. The same

regarding the neutron reflectometry measurements; the extent of the changes in the acquired reflectometry data is the same as that typically used in many cases in literature. The changes are small but certainly significant considering the sensitivity of the techniques based on short-range order compared to long range XRD.

Neutron reflectometry and XAS are sensitive enough techniques to monitor the small changes as shown in the submitted manuscript and for reference, just a few of many possible examples already found in literature:

- Hiayama, M., Shibusawa, T., Yamaguchi, R., Kim, K., Taminato, S., Yamada, N., Kanno, R. (2016). Neutron reflectometry analysis of $\text{Li}_4\text{Ti}_5\text{O}_{12}$ /organic electrolyte interfaces: characterization of surface structure changes and lithium intercalation properties. *Journal of Material Research*, 31(20), 3142-3150. Doi: 10.1557/jmr.2016.320.
- Gilbert, D. A. *et al.* Controllable positive exchange bias via redox-driven oxygen migration. *Nat. Commun.* 7:11050 doi: 10.1038/ncomms11050 (2016).
- Kaur A, Singh A, Singh L, Mishra SK, Babu PD, Asokan K, *et al.* Structural, magnetic and electronic properties of iron doped barium strontium titanate. *RSC Advances* 2016, 6(113): 112363-112369.
- Upton MH, Choi Y, Park H, Liu J, Meyers D, Chakhalian J, *et al.* Novel Electronic Behavior Driving NdNiO_3 Metal-Insulator Transition. *Physical Review Letters* 2015, 115(3).
- Huang Z, Raghuwanshi VS, Garnier G. Functionality of Immunoglobulin G and Immunoglobulin M Antibody Physisorbed on Cellulosic Films. *Front Bioeng Biotechnol* 2017, 5: 41.

Regarding the reproducibility of the measurements, we used three independent sets of samples that were characterised in three dedicated measurement campaigns at the synchrotron light source and two at the neutron source. The best data was acquired with the third set of samples after optimisation of the experimental setups, but all three sets showed clearly the exact same effects. This information is now provided in the experimental section of the revised version of our manuscript where we added the following in the experimental section at lines 438-441:

“In this work we used three independent sets of samples that were characterised in three dedicated measurement campaigns at the synchrotron light source and two at the neutron source. The best data was acquired with the third set of samples after optimisation of the experimental setups”.

Reviewer 3 point 2:

To date there have been no examples of possibilities to grow stoichiometric LaTiO_2N thin films. We have not been able to grow films with substitution higher than that of $\text{O}_{2.2}\text{N}_{0.8}$ using Pulsed Reactive Crossed-beam Laser Ablation (refer to experimental section). Previous works have extensively explored the growth of these oxynitrides and shown that it is not possible to grow oxynitrides with N content higher than a few percentage by conventional Pulsed Laser Deposition.

- Pichler M, Pergolesi D, Landsmann S, Chawla V, Michler J, Döbeli M, *et al.* TiN-buffered substrates for photoelectrochemical measurements of oxynitride thin films. *Applied Surface Science* 2016, 369: 67-75.

We also observed the trade off between crystalline quality and nitrogen content, where films with increased nitrogen contents typically exhibit highly textured or polycrystalline structures. Whereas all epitaxial films have O/N ratios of around 10%. We would like to point out that for the composition analysis (RBS and ERDA) the relative error of the ratio is about 7%. The relative uncertainty of the absolute stoichiometric coefficient of N is actually about 10% but there the uncertainty of oxygen is included which cancels out when giving the O/N ratio. As we have already referred to in the supplementary.

The only other example of LTON thin films found in literature is listed here:

- Le Paven-Thivet C, Ishikawa A, Ziani A, Le Gendre L, Yoshida M, Kubota J, *et al.* Photoelectrochemical Properties of Crystalline Perovskite Lanthanum Titanium Oxynitride Films under Visible Light. *J Phys Chem C* 2009, 113(15): 6156-6162.

Here the authors only report relative O/N ratios determined by EDX analysis and do not provide the overall compositions of their samples and are referred to as LaTiOxNy and LTON.

We do agree with the reviewers comments that a further study comparing epitaxial, textured and polycrystalline thin films using NR and XAS would be of interest. This investigation would make it possible to compare the effects of different crystallinity, N content, and morphology. However, that would require additional beam-time allocations (in a very significant amount) at a synchrotron source and neutron source. With the present study we set the basis for our future work and what the reviewer suggests is part of our future research interests.

The remark raised by reviewer 2 and highlighted by reviewer 3 about the non-stoichiometric N content definitely deserves a more detailed discussion. Our previous answer to this point was evidently not convincing enough. Reviewer 3 asks:

“Do the cell parameters emphasize a (strong, weak) distortion of the perovskite ABO₂N cell and how does this influence the behaviour of A and B cations during the photoelectrochemical test?”

The LTON crystal structure remains the same, orthorhombic perovskite, within a large range of N content. The cell parameter, in turn, can significantly be affected by the N content because the N content affects the Ti – O – Ti bond angle and the overall electronegativity of the anions, i.e. ultimately the overall distortion of the cell. In literature, it is well accepted that the Ti – O – Ti bond angle and the anion electronegativity affect the electronic band structure. The first parameter changes the width of the conduction band leaving the energetic centre of the band unaffected. The second influences the energetic centre of the conduction band but not the width [1].

[1] Aguiar R, Logvinovich D, Weidenkaff A, Rachel A, Reller A, Ebbinghaus SG. The vast colour spectrum of ternary metal oxynitride pigments. *Dyes and Pigments* 2008, 76(1): 70-75.

Both these features have an important influence on the light absorption properties because the overall band gap changes. However, the light absorption properties refer to the bulk, where we did

not observe any significant changes comparing before and after PEC tests. We observed changes of the electronic and geometric structure only at the surface and only after the oxygen reduction reaction took place (applied bias plus light illumination). We have no control on the exact chemical composition of the utmost layers, independently on the chemical composition of the bulk. The use of the stoichiometric composition LaTiO_2N , or the composition we could experimentally obtain in our thin films, it is not a critical parameter because also in the case of the stoichiometric composition of the sample, the surface composition can be different from that of the bulk depending on fabrication method and sample history. The specific value of N content may have an influence on the overall extent and velocity of the observed effect because the initial chemical composition of the surface is different. However, we cannot conceive of any reason why the basic mechanism could be different and/or so dramatically dependent on the N content of the films. We acknowledge that this is an assumption. Although the most likely, logical and straightforward, but still an assumption that at the current stage cannot be proven. For this reason, following the reviewer's concern, we add the following sentence in the revised version at lines 97-99:

"We assume that the chemical composition of the films, while affecting the light absorption and charge migration properties, has little influence on the evolution of the physicochemical state of the surface layer."

Reviewer 3 point 3:

We personally agree here with the reviewer comment that the pre edge section of the supplementary is of interest and detailed discussion and we did consider to include the relevant section in the main text of the manuscript.

However, we have also received other points of views that for a broader readership such as that for nature communications, that section may be too detailed and too specific for a general audience. We also considered and we appreciate that publication space is limited and valuable. Therefore, we were a little conscious that the manuscript may be too long if included in the main text.

On balance, we feel that the whether the section is omitted or not, the overall understanding of the manuscript is not affected. Therefore, we chose to err on the side of caution and reference the relevant section in the main text and people can refer to as and when interested.

From our side, both options to include/exclude the pre-edge discussion in the main text are acceptable. We defer to the editors for their suggestion on this point before revising the manuscript further.

Best regards,

Craig Lawley